# TRANSNETS FOR REVIEW GENERATION

**Rose Catherine & William Cohen**
School of Computer Science
Carnegie Mellon University
{rosecatherinek, wcohen}@cs.cmu.edu

## ABSTRACT

In recommender systems, review generation is increasingly becoming an important task. Previously proposed neural models concatenate the user and item information to each timestep of an RNN to steer it towards generating their specific review. In this paper, we show how a student-teacher like architecture can be used to rapidly build a review generator with a low perplexity score.

## 1 INTRODUCTION

Personalized Recommender Systems need to not only produce good recommendations that suit the taste of each user but also provide an explanation that shows why a recommendation would be interesting or useful to the user, to be more effective [Tintarev & Masthoff (2011)]. Showing a user how their review would look like for an item if they were to try it is an important step towards generating a detailed personalized explanation. For this reason, review generation is gaining popularity in the recommendation community. It is a type of controlled text generation task, where the generated review should highlight the aspects of the item this particular user would like (or dislike). Therefore, it is influenced by both the user and the item.

RNNs have been shown to be very good at modeling language [Zaremba et al. (2014)]. Once trained, they can generate samples from the language model. Controlling the content that is generated is usually achieved by providing a context vector to the RNN. A recent work called Collaborative Filtering with Generative Concatenative Networks (CF-GCN) [Ni et al. (2017)] proposed to concatenate latent embeddings of a user-item pair to the input in each timestep of the RNN in an effort to force it to generate their joint review. Albeit a simple strategy, the authors showed that by providing the user-item information to the RNN at each step, they could achieve very low perplexity scores.

In this paper, we show how a student-teacher like architecture called ***Transformational Neural Networks*** (TransNets) proposed by Catherine & Cohen (2017) for the task of rating prediction can be used for review generation. By introducing an additional latent layer that represents the review of a user for an item, which at training time is regularized to be similar to the actual review's latent representation, they showed that they could improve the prediction performance substantially. In this paper, we show that such an intermediate step also dramatically improves the performance of a review generator as well. Our experiments show that such an approach achieves a low perplexity within a single epoch, which CF-GCN takes 6 epochs to achieve. That corresponds to about 4x speedup in training time.

## 2 PROPOSED APPROACH: TRANSNETS FOR REVIEW GENERATION

In our proposed architecture for review generation, there are two networks as shown in Figure 1. A *Target Network* (teacher) processes the text of $user_A$'s review for $item_B$, denoted as $rev_{AB}$, using a text processor $\Gamma_T$, which embeds the words and extracts sentence or phrase level features, to construct a latent representation for the review. This latent vector is then concatenated to each step of a decoder LSTM to reconstruct $rev_{AB}$. Therefore, the Target network is essentially, an autoencoder. By training the Target Network in this manner, it learns to generate a good encoding of the text to minimize the reconstruction error. Also, this network achieves a low perplexity without much difficulty because it is working with the actual review.

The proposed architecture has a second network called *Source Network* (student) that first embeds $user_A$ and $item_B$ using embedding matrices $\Omega_A$ and $\Omega_B$, and subsequently, learns how to trans-

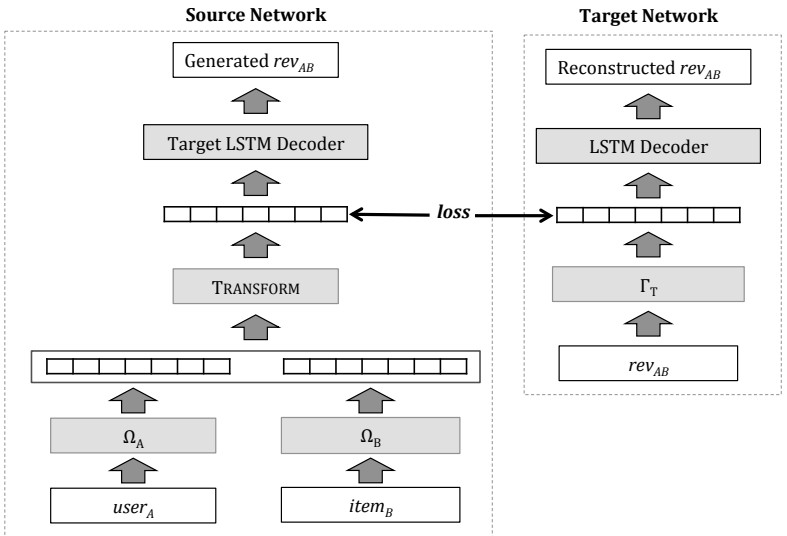

Figure 1: TRANSNET Architecture adapted for Review Generation

Table 1: Dataset Statistics

| Dataset | #Users | #Items | #Reviews | Train | Validation & Test | Avg. Review Length |
|---------|--------|--------|----------|-------|-------------------|--------------------|
| BA_150 | 1,601 | 1,615 | 541,996 | 538,744 | 1,626 each | 123 words |

form the two latent representations into that of their joint review, $rev_{AB}$. This is achieved by using a TRANSFORM layer, which is a L-layer deep non-linear fully connected feed forward network. During training, we will force this layer to produce a representation that approximates the Target Network's encoding for $rev_{AB}$, by minimizing the $L_2$ loss computed between the two latent representations. At test time, for a test user-item pair, $user_P$ and $item_Q$, the Source Network is used to construct an approximate representation of their joint interaction, which is fed as a context vector to the decoder LSTM that was trained by the Target Network, to generate their joint review.

Although TransNets have teacher-like and student-like networks, there are important differences between the two. We refer interested readers to Catherine & Cohen (2017). Similar to the CF-GCN model [Ni et al. (2017)] discussed above, Tang et al. (2016) produce the text of the review from the item id and sentiment/rating as input. Wang & Zhang (2017) applied a Dynamic Memory Network (DMN) to generate a personalized latent representation of the joint review using user's latent vector as attention, which is then passed through an LSTM to generate the review. Li et al. (2017) is another recent model that generates short tips instead of the full review using an RNN.

## 3 EXPERIMENTS AND RESULTS

To study the effectiveness of our proposed approach, we used a beer review dataset called `Beer Advocate` [McAuley et al. (2012)]. It has 33,387 users, 66,051 beers, and 1,586,259 reviews and ratings. For our preliminary studies, we created a *k-core* subset of this dataset, where $k$ is 150. A *k-core* is the largest possible subset of users and items such that each user in the subset has written reviews about at least $k$ items in the *k-core* subset and each item in the subset has at least $k$ reviews written about it by users in the subset. The statistics of this 150-core dataset, henceforth referred to as `BA_150`, are given in Table 1. For each user, we randomly sample two of their interactions as their validation and test cases, and use the remaining interactions for training (leave-one-out evaluation). We use the vocabulary of 20,000 most frequent words in the train dataset. We fixed the maximum length of reviews to 100 tokens. Smaller reviews were padded with 0's.

### 3.1 EVALUATION PROCEDURE AND SETTINGS

For the Target Network's encoder, we use a CNN Text Processor with the same parameter settings as Catherine & Cohen (2017), and producing a 100D representation for the text. In the student network, both users and items are embedded into a 50D latent space. The TRANSFORM module is 2 levels deep, and transforms the user and items embeddings into a 100D latent space. In the first epoch, for the first 500 mini-batches, the Source network is not trained; Only the Target is trained. After that, both networks are trained in each of the mini-batches.

The base language model is TensorFlow's benchmark implementation[1] of Zaremba et al. (2014) which achieves very good results for language modeling tasks. We use their recommended `medium` configuration: 2 layers of stacked LSTMs with 650 hidden units, but with 100 timesteps instead of their 35, and a mini-batch size of 40. The network is regularized using dropouts (0.5 keep probability). The prior state-of-the-art, CF-GCN, was trained and tested using the same settings as above.

### 3.2 EVALUATION

Our metric for performance evaluation is word level perplexity (lower is better). After each epoch (one iteration through the whole training data), we measure the perplexity of the models on the validation and test sets. We chart the perplexity obtained on the test dataset in Figure 2. Perplexity on validation data follows the same trend.

It can be seen from the figure that with just one iteration over the training data, the TransNet's Source (student) network has achieved a low perplexity of 34.095, closely following TransNet's Target network, which is at 31.845. This shows that the additional TRANSFORM

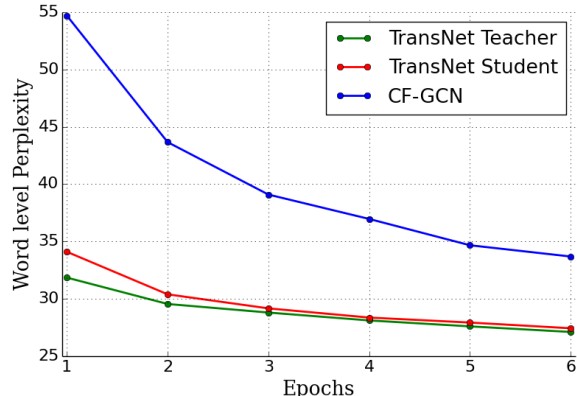

Figure 2: Test set perplexity as training proceeds

layer is quite successful in constructing an approximate representation that can be fed into the Target's LSTM decoder. CF-GCN starts at a perplexity of 54.696 after 1 epoch and takes 6 epochs to reach the same level of performance as TransNet's first epoch. CF-GCN takes about 1.5 hours per epoch on a NVIDIA GeForce GTX TITAN X GPU to train, and TransNet takes 2 hours per epoch. That corresponds to about 4x improvement in training time for the same level of performance. It shows how we could use the TransNet framework to rapidly train a review generator and it will be beneficial in many real world settings while training on large datasets.

A sample produced by the Source network after 1 epoch of training: *Appearance: Pours a dark black color with a small head that disappears quickly. Smell: The aroma is of coffee, roasted malts, and a hint of coffee. Taste: The taste is very bitter with a nice roasted malt flavor. The chocolate really comes through in the finish. Mouthfeel: This beer has a medium body with moderate carbonation to it. Drinkability: This is a very drinkable beer. I could drink this all day long.*

## 4 CONCLUSIONS

In this paper, we showed that by using a student-teacher like architecture called TransNets, it is possible to train a good quality review generator in a very short time. Our model used a Target Network that learned how to produce good internal representations of a review text so as to be able to reconstruct the original review. A Source Network learned to embed users and items, and to combine the two latent representations to mimic the internal representation produced by the Target for a user-item pair. Our preliminary experiments show the efficacy of our proposed approach.

---

[1]https://github.com/tensorflow/models/tree/master/tutorials/rnn/ptb

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
