# OpenReview forum: "TransNets for Review Generation"
_ICLR.cc/2018/Workshop — Accept_

### Official Review · AnonReviewer2 · 2018-03-05
**Shows that performance of CF-GCN can be improved via various architectural changes; an effective if not huge contribution.**

**Rating:** 7
**Confidence:** 4

**Review:**

The paper looks at new models for review generation, i.e., the problem of generating a review that matches a given (user,item) pair.

The main insight is to adapt a recent model known as the student/teacher architecture for this problem.

This is a pretty reasonable solution, and suitable for a workshop paper. Essentially the authors show that a specific architectural change can improve model performance somewhat, which is a pretty standard if not huge contribution.

Some overall comments about the paper:
- Not clear if training really proceeds to convergence in Figure 2 or whether all methods wouldn't benefit from additional training
- Not clear whether there's a qualitative difference between the reviews generated by various methods, something which (e.g.) a user study might help with
- Not clear how to address scalability issues (to larger/sparser datasets with many users/items), which was a large focus of the CF-GCN paper

Other than that the technique seems reasonable and effective

---

### Official Review · AnonReviewer1 · 2018-03-07
**An interesting approach**

**Rating:** 6
**Confidence:** 4

**Review:**

This paper considers the task of "review generation". The motivation is recommender systems: it is often desirable to generate reviews that a user might be expected to write for a recommended item. For this task the authors propose to harness an architecture that they recently proposed in prior work: TransNets. The idea is to incorporate a latent layer that is meant to represent the review a user *would* write for a given item.

More concretely, TransNet uses an auto-encoder approach to train a "teacher" (or target) network learn to induce representations of reviews of an item B by user A, x_{AB}. A source network ("student") is then trained to estimate these vectors, given only user and item representations. This is a neat approach, although not novel to the present work.

It seems like a downside to TransNets is the need for a fairly large set of reviews written by any given user to train a reasonable auto-encoder (teacher network). That the authors constructed a "150-core" dataset, ie., a dataset in which all users have written about 150 items, and all items have reviews written by at least 150 items. This seems quite restrictive: what fraction of users have written this many reviews, one wonders?

Despite this practical weakness, I think the idea is interesting and this may facilitate further discussion at the workshop.

---

### Official Review · AnonReviewer3 · 2018-03-09
**Interesting architecture**

**Rating:** 8
**Confidence:** 4

**Review:**


The authors propose to use a TransNet architecture (recommender system framework relying mainly on textual data) to generate reviews. The idea is to exploit the original architecture, based on two parallel networks, to build a relevant joint profile associated to both an item and a user. Then, the authors propose to learn a LSTM based decoder to generate a review (from this profile).
They demonstrate the interest of their approach wrt CF-GCN according to the perplexity measure.

The article is well written and the architecture interesting. This formulation is an elegant way to tackle the black-box phenomenon around recommender systems. For all those reasons, I think that this article should be presented in ICLR.

However, I have a few more questions:
- the author claim that their architecture is more efficient... But we would like to see how both models behave with more epochs... Given the slopes of the plots, we can imagine that all performances will converge quickly.
- the discussion around teacher-student networks seems quite unfair: it is only a matter of supervision in this article. The architecture requires to perform the back-propagation from supervised data that are mapped into the latent space... Their is no direct link with teacher-student architectures which aim at approximating a compex process with a lighter network.
- The task is very difficult to evaluate... It would be interesting to add basic baseline to provide a better understanding of the performance. For instance, we could add a random baseline (with words drawn from the general distribution of the item review) or a best-review baseline measuring the perplexity wrt the best existing review in the learning set (accroding to the target user).
One major issue is to demonstrate that the generated reviews are more interesting that reviews extracted from the dataset.
cf Extended Recommendation Framework: Generating the Text of a User Review as a Personalized Summary, Poussevin et al. Workshop on New Trends in Content-Based Recommender Systems, RecSys 2015

---

### Decision · Program_Chairs · 2018-03-20
**ICLR 2018 Workshop Acceptance Decision**

**Decision:**

Accept

**Comment:**

Congratulations, your paper was accepted to the ICLR workshop.